# Arduino Automated Microwave Oven for Tissue Decalcification

**DOI:** 10.3390/bioengineering10010079

**Published:** 2023-01-06

**Authors:** Paolo Savadori, Sophia Dalfino, Marco Piazzoni, Francesco Inchingolo, Massimo Del Fabbro, Gianluca Martino Tartaglia, Luciano Giardino

**Affiliations:** 1Department of Biomedical, Surgical and Dental Sciences, Università degli Studi di Milano, 20133 Milan, Italy; 2IRCCS Fondazione Ca’Granda IRCCS Ospedale Maggiore Policlinico, 20122 Milan, Italy; 3Complex Tissue Regeneration Department, MERLN Institute for Technology-Inspired Regenerative Medicine, 6229 Maastricht, The Netherlands; 4Advanced Biomaterials Lab, Physics Department, Università degli Studi di Milano, 20133 Milan, Italy; 5Interdisciplinary Department of Medicine, Università di Bari “Aldo Moro”, 70121 Bari, Italy; 6Independent Researcher, 88900 Crotone, Italy

**Keywords:** Arduino, bone, decalcification, microwaves, teeth

## Abstract

Decalcification of hard tissues such as bone and teeth is a complex process that requires using chemicals such as acids and chelating agents. Acids act faster than chelating agents, but they have a greater risk of damaging biological samples. Increasing the reaction speed of the chelating agent may solve this issue. There are several strategies to speed up this process, and using microwaves seems to be one of the most effective. However, lab-dedicated microwave ovens are expensive, and their purchase may seem unjustified. Therefore, a low-cost modification of a commercial microwave oven, consisting of an Arduino automation device, has been developed. The setup has proven reliable for continuous work, thanks to implementing an electronic safety circuit. In addition, it may reduce the decalcification time using a chelating agent, achieving optimal results regarding tissue preservation and quality of histological sections.

## 1. Introduction

Hard tissues, such as bone and teeth, must undergo a decalcification process to obtain a histological section. Decalcification is a crucial point; if it is not performed correctly, it will render the sample unusable. Many approaches have been proposed through the decades, but they all can be divided according to the chemicals used: acids and chelating agents. Acids carry out their activity by dissolving calcium deposits and forming carbon dioxide and salts, which differ according to the acid used [1,2]. A chelating agent acts by binding the single calcium ion or divalent ions [3]. The main differences between these approaches are the speed and the amount of tissue damage, the latter being proportional to speed. Acids can achieve complete decalcification relatively quickly, but they often cause consistent tissue damage due to the low pH value.

On the other hand, chelating agents, such as EDTA, can work at physiological pH, avoiding tissue damage. Still, they may need up to several weeks or months to completely decalcify a bone or tooth specimen. Nevertheless, keeping a sample in a chelating agent for too much time could damage the tissue by deteriorating collagen and other protein structures [4].

It is possible to try to speed up the process and preserve the structure of the sample from acidic damage. For example, substances such as formalin could be added to an acid solution to fix the specimen and reduce tissue damage [5]; a buffered acid solution can raise the pH of an acidic substance, partially extending the decalcification time but lowering tissue damage. Constant agitation of the biological piece may accelerate the process independently from the solution used. In addition, ion exchange resins reduce decalcification time by sequestering calcium ions from the decalcifying solution [6,7]. Another method to increase the decalcification speed is to warm up acids or chelating agents. While this process increases the reaction kinetics, it could severely damage the sample if it is not carefully monitored.

Microwave radiation is another technique that can speed up decalcification, reducing the chance of causing tissue damage [8]. Microwaves react with polar liquids by changing the orientation of the molecules. This motion increases the chemical reactivity at the cost of a temperature rise. The main drawback of professional microwave ovens is the relatively high cost. In a previous investigation, decalcification was carried out using a commercial microwave oven [9]. Samples submerged in EDTA are irradiated for a specific duty cycle to prevent the solution from reaching a harmful temperature. Still, the oven must be controlled by personnel, so having a 24-h continuous duty cycle becomes unpractical.

This study aimed to develop a method to automate a low-end commercial microwave through an Arduino board and a few essential electrical components. This setup will include a safety circuit to avoid any issues derived from crashes in the Arduino’s software (Arduino IDE 1.8.19, Milan, Italy) that could alter the oven’s working cycles.

## 2. Materials and Methods

The following materials were used to assemble the entire instrument; in Table 1 are reported prices and stores:-A commercial oven with mechanical input commands (SilverCrest SMW 700 B1, Kpmpernass GMBH, Bochum, Germany). Ovens with electronic timers are unsuitable because bypassing the installed circuit boards is difficult.-An Arduino board: Arduino is an open-source electronic microcontroller, and there is a large variety of models on the market. Each model works similarly, but it was decided to use the original Arduino UNO (Interaction Design Institute, Ivrea, Italy).-Two 100 nF capacitors, one resistance of 10 Kohm, one of 3300 Ohm, two diodes model 1N4140, one transistor NPN model 2N2222 (or NPN BC 547b).-Breadboard and connection cables-One Arduino-compatible mechanical relay-An array of neon mini lamp bulbs

**Table 1 bioengineering-10-00079-t001:** Price list of electronic components.

Component	Store	Price
Arduino	Arduino official store	23 $
Capacitors	Digikey.com	0.92 $
Resistance	Digickey.com	0.60 $
Dioedes	RS-online.com	1 $
Breadboard and connection cables	Digikey.com	23 $
Transistor	RS-online.com	0.27 $
Microwave oven	Lidl Italy SRL	65 $
Neon bulbs	Farrel.it	103 $
Total		226,84 $

### 2.1. Microwave Oven Modifications and Settings

Commercial microwave ovens suffer from a high level of unbalanced microwave irradiation. Therefore, it is crucial to identify hot and cold spots inside the oven chamber to address this problem. Login and Dovorak set up a procedure to map the irradiation pattern inside a microwave oven [10]; this technique requires an array of many led-sized neon bulbs and a container, such as a 96 multi-well plate. When enough energy strikes a neon lamp, it will emit light, pointing to that region as a hot spot and a possible place to put a sample.

Usually, mechanically controlled ovens have one knob to adjust the power and the other for time settings. Power must be set at maximum value. In this case, the maximum power output was 700 watts. There are two options for the timer: bypassing the temporized switch and substituting it with a conventional on-off switch or resetting the timer from time to time.

The mechanical relay must be connected to the phase wire of the oven (usually the brown cable inside the oven power cord). Next, the relay control pin must be plugged into the safety circuit board, which is connected to a digital pin on the Arduino board. Finally, the relay must be set on “normally open” (NO): in this way, when the relay is not powered up, it will remain open, not allowing the passage of electricity, and keeping the oven switched off.

### 2.2. Safety Electronic Circuit Board

This microwave oven setup aims to work 24 h a day. For this reason, some safety precautions are needed. Managing a relay with Arduino is very simple and consists of a single electric signal that opens or closes the circuit inside the relay. However, this mechanism is prone to possible malfunctions, such as a software crash that can leave the relay circuit closed, leading to continuous irradiation with the consequent sample destruction. To overcome this risk, the relay may operate through a square wave impulse generated by the Arduino board, rectified by the safety electronic circuit board. The square wave is produced by a rapid switch on and off the Arduino pin’s electric signal (Appendix A). However, the safety circuit converts this pulsed signal into a continuous one. Therefore, the rectifier board needs an intermittent signal to generate a steady impulse. In the eventuality that Arduino is not capable of generating the square wave, e.g., the software stops at some point during the cycle, the electric signal is not delivered to the relay, and the microwave automatically shuts down. The safety circuit diagram is reported in Figure 1, and the complete setup scheme is summarized in Figure 2.

A second microwave oven with only the Arduino ONE board and the mechanical relay was used as a control, running the same duty cycle as the one with the auxiliary circuit, to see the actual effect of the control device. Signal cable plugged in pin 3, 5v red cable plugged in 5V pin, black ground cable plugged in GND pin.

### 2.3. Sample Placement, Decalcification, and Duty Cycle

Hard tissue samples were processed to test the modified microwave’s correct working. Almost any decalcifying solution can be used in the microwave oven. However, some chemicals could be dangerous when heated up, potentially releasing toxic vapors. For this study, 14% EDTA disodium salt at pH = 10 was chosen. EDTA is a well-known chelating agent with relatively slow action and does not alter tissue morphology [11]. The same kind of tissues was decalcified as control with Anna Morse’s (AM) solution at room temperature, following the standard protocol [12].

The control group included samples decalcified in AM solution, and the test group included samples decalcified with EDTA in the microwave oven (EDTA-MW). Each group was composed of 3 sections of cow molar teeth of the same weight of 2 g and with periodontal tissue, 3 mouse vertebrae, and 3 mouse heads. Cow teeth were taken from animals for human consumption, while mouse tissues were taken from discarded material from other studies. For these reasons, no ethical committee was needed. Hard tissue samples must be put in a plastic container (plastic materials are permeable to microwave radiation), and filled with a decalcifying solution. The sample–liquid ratio has to be 1:50 in volume. Biological samples were irradiated 10 s per hour, 24 h a day, until complete decalcification. The temperature was checked with an LLG pH-PEN (Lab Logistics Group, Meckenheim, Germany) after each cycle for 8 h during the first working day to ensure that the temperature did not exceed the critical level of 50 °C; this temperature was not reached. All decalcifying solutions were replaced every 48 h. The decalcification endpoint was assessed through a physical test by bending and piercing the sample in a non-point-of-interest zone [13]. The sample was considered decalcified if a needle could pass through the hard tissue without encountering any resistance. In addition, two cow’s teeth were left in the decalcification medium and microwave irradiation for 90 days after the complete decalcification to assess tissue damage due to over-treatment.

### 2.4. Histological Analysis

Decalcified tissue samples were dehydrated through a series of ethanol solutions from 70% to 100%, cleared in xylene, and then embedded in paraffin. Sections of 5 μm were obtained with a rotatory microtome (Leica RM 2245 rotatory microtome, Leica Biosystems Nussloch GmbH, Nussloch, Germany) and put on a polilysinated glass slide (VWR International, Radnor, PA, USA). Paraffin sections were stained with trichromic eosin made by eosin, phloxine B, and Orange G and counterstained with Morel & Bassal iron hematoxylin [14]. Eosin and phloxin B combination is widely used. Still, Orange G was used to highlight the collagen structure [15] better. All chemicals were purchased from Carlo Erba Reagents S.r.l. (Milano, Italy). The staining procedure is described in Appendix A. Microphotographs were taken with a CX43 Olympus microscope and an LC30 Olympus digital camera (Olympus Italia, Segrate, Italy).

## 3. Results

The modified microwave oven with the auxiliary board worked flawlessly for over 90 days, counting the over-treatment time. However, the microwave without the safety circuit experienced a catastrophic failure after 30 days of work: the relay did not receive the opening signal preventing the oven’s shutting down, which eventually shut off thanks to the mechanical timer. This test has no statistical relevance but has shown that the Arduino board could be prone to failure.

Cow’s teeth in AM solution took 15 days to complete decalcification, whereas mouse head and vertebrae took 5 days. Cow’s teeth in EDTA-MW took 25 days, whereas mouse head and vertebrae took 14 days. AM solution is faster than EDTA, but the microwave can speed up the decalcification process with the chelating agent from months to weeks. The tissue from the cow’s teeth and periodontal tissue were well preserved independently of the method used. Still, samples decalcified with EDTA-MW showed slightly better stainability, especially the periodontal tissue. Teeth processed for 90 days after the decalcification endpoint did not show any tissue damage, and they were comparable to the samples processed for the needed time (Figure 3).

Mice heads and vertebrae processed with AM solution showed tissue damage at the central nervous system level and worse bone marrow stainability (Figure 4) compared to test samples, which did not show signs of damage to the CNS. The damage can be attributed to the low pH of the AM solution.

## 4. Discussion

The optimal decalcification of a mineralized tissue implies a delicate balance between time and tissue damage. For example, strong mineral acids can completely decalcify hard tissues in a few days, while chelating agents could take several weeks or months in the case of large samples or highly calcified tissues such as teeth [16]. However, the speed of this process is at the cost of sample quality, and strong acids cause various types of damage, from structural alteration to staining issues. An intermediate alternative is using organic or buffered organic acids such as AM. These solutions can allow complete decalcification in a reasonable amount of time while preserving good quality for most tissues. Anyway, delicate tissue structures such as the central nervous system could sustain damage, and the low pH could alter the protein structure, causing difficulties in the histochemical staining technique [17].

Chelating agents offer the best tissue preservation at the cost of a very long procedure time. Therefore, speeding up the kinetics of this class of chemicals may allow for achieving optimal results in a short time. This could be achieved through different strategies, such as heating the liquid, applying constant agitation, or using ion exchange resins. However, all these approaches have some drawbacks, or a marginal benefit is derived [18,19]. 

Microwaves can solve this problem thanks to their function: microwave radiation interacts with polar molecules, such as water, rapidly changing their orientation [20]. This motion causes a heat rise and increases the kinetics of the chemical reaction. Furthermore, microwaves can enter in resonance with the crystalline structure of hydroxyapatite, causing destabilization and facilitating the interaction between the decalcifying solution and bone [21].

Expensive lab-oriented microwave ovens can perform many functions other than decalcification and, in general, are complex machines. However, if there is only the need to process calcified tissue, it is reasonable to use a modified commercial oven. Moreover, implementing automatization with an Arduino board can create an instrument that can be used in laboratory routines [22]. Therefore, the setup reported in this work is low-cost, and the safety circuit allows one to put on work 24 h a day and needs a minor human operation that can be summarized in changing the decalcification medium and resetting the oven’s timer if needed.

The most critical aspect is that the temperature must be kept below 50 °C to avoid heat tissue damage [23]. Therefore, the duty cycle for this work was set at 10 s per hour. Still, this setting could change depending on the volume of the decalcifying solution: a 1:30 sample/solution volume ratio is the minimum quantity for a correct decalcification, but the solution volume could be increased [24]. Therefore, the sample could be irradiated for an extended time. Nevertheless, the cooling time must be considered to have the best balance between treatment time, cooling, and total volume.

Decalcifying agents of any kind could be used with this methodology but with due precaution: strong mineral acids could release toxic vapor such as chlorine or nitrogen oxide. For this reason, it is advisable to use the oven under a chemical hood in case hazardous substances are used. This study used 14% disodium EDTA at pH 7, a standard chelating solution for decalcification, and the samples obtained were compared to other samples of the same type, decalcified with AM solution. This correlation aimed to evaluate the tissue quality after the microwave process and assess the timing of the procedures. However, the small sample size cannot be considered statistically relevant, and it was intended to be a qualitative (proof-of-principle) survey.

Both AM solution and EDTA-MW show an optimal result concerning cows’ teeth decalcification and periodontal tissue preservation: the time needed for EDTA solution to achieve a complete decalcification was 10 days more than AM solution; for this reason, in the case of highly calcified tissued it is more convenient use an organic acid. There is no substantial difference in staining between the two processed tissues except for a slightly darker color in the keratinized mucosa of the cow’s periodontal tissue (Figure 3). This change in color tone could be attributed to more acidity of the stained section [25]. Odontoblasts were attached to the canal walls and showed good nuclei staining; the pulp preserved its structure without signs of damage. Gingiva was well characterized with good differentiation between the pluristratified epithelium and the connective tissue below and marked nuclei. This differentiation has been emphasized by adding orange G to the eosin mix. Orange G is used in trichrome staining for collagen, such as Mallory’s staining. Its proprieties could be added to the hematoxylin and eosin staining for a more efficient tissue architecture evaluation.

AM solution shows some limitations in bone marrow staining and preservation: bone marrow quantity in vertebrae appears less in AM decalcified samples than in EDTA-MW decalcified ones. Furthermore, bone marrow was darker in the specimens treated with the acid, in contrast to the ones decalcified with the chelating agent. This can be attributed to the difference in pH (Figure 4). Additionally, the cerebellum and spinal cord showed a high degree of damage in the samples processed with AM compared to the pieces processed with EDTA-MW. Finally, cow’s teeth with periodontal tissues treated for 90 days showed no abnormality or staining artifacts. These data indicate that using microwave and EDTA could be helpful in all the cases that are needed to decalcify composite tissue, such as vertebrae, articulation with bone and muscle attached, mouse or rat heads [26], and teeth with periapical or endodontic lesions [27]. Furthermore, this process could be applied when there is a necessity for decalcified biopsies of regenerated bone which are used as substitutes different from the natural bone-like dentin: it is possible to prolong the decalcification time to be sure that the different types of calcified tissues have reached their endpoint without the worry of over-process time [28,29]. The same principle could be applied in decalcifying a rodent’s jaw or mandible: bone will reach the decalcification endpoint much sooner than teeth, but the prolonged processing time wo not affect bone, or soft tissue, architecture.

## 5. Conclusions

Microwaves could aid the decalcification procedures, speeding up the chelating agent activity, even if this may not significantly reduce the total time needed for achieving tissue decalcification. Furthermore, they can be helpful in decalcifying composite tissues without caring about over-processing.

The introduction of Arduino boards has allowed a relatively easy way to implement the automatization into various instruments and machines that lack this feature. In this work, we used the potentiality of this microcontroller to transform a commercial microwave oven into a lab instrument. Furthermore, a safety control circuit has been introduced to increase reliability and allow continuous functioning. Finally, with this setup is possible to spare a considerable amount of resources while still having an instrument with similar capability as an expensive lab counterpart.

## Figures and Tables

**Figure 1 bioengineering-10-00079-f001:**
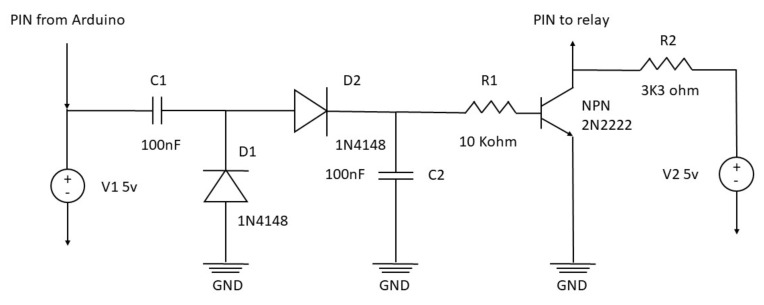
Electric diagram of the safety control board.

**Figure 2 bioengineering-10-00079-f002:**
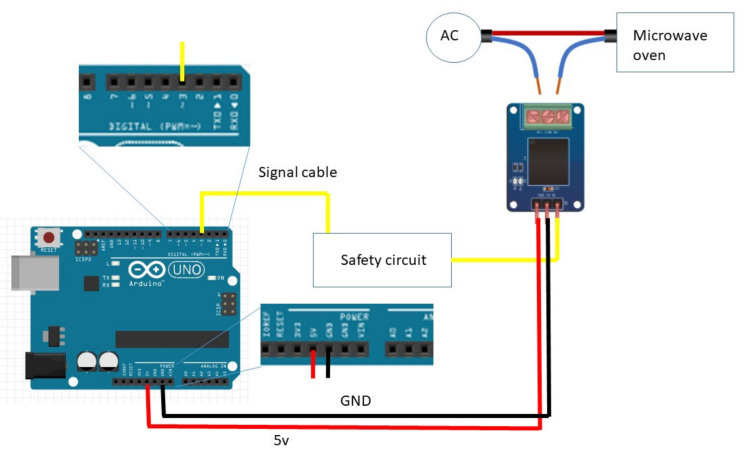
General connections of the complete setup.

**Figure 3 bioengineering-10-00079-f003:**
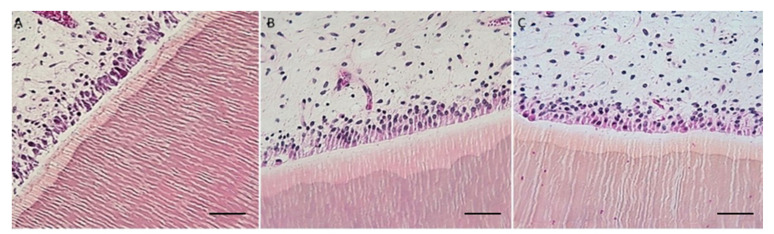
Cows teeth and periodontal tissue. (**A**) Cow tooth decalcified with Anna Morse’s solution, 100× scale bare 100 µm H&E staining. (**B**) Cow tooth decalcified with EDTA-MW, 100× scale bare 100 µm H&E staining. (**C**) Cow tooth decalcified with EDTA-MW for 90 days, scale bare 100 µm H&E staining. It is possible to recognize pulp tissue in the upper part of the images, odontoblast attached to the canal walls, and dentin with its characteristic dentin tubules. Dentin is composed of pre-dentin, a lighter shade of pink, and dentin, a darker shade of pink. (**D**) Periodontal tissue treated with Anna Morse’s solution scale bare 100 µm H&E staining. (**E**) Periodontal tissue treated with EDTA-MW scale bar 100 µm H&E staining. (**F**) Periodontal tissue treated with EDTA-MW for 90 days scale bare 100 µm H&E staining. It is possible to distinguish the pluristratified keratine cell layer in the upper part of the images, stained in purple, and the connective tissue below dyed in orange.

**Figure 4 bioengineering-10-00079-f004:**
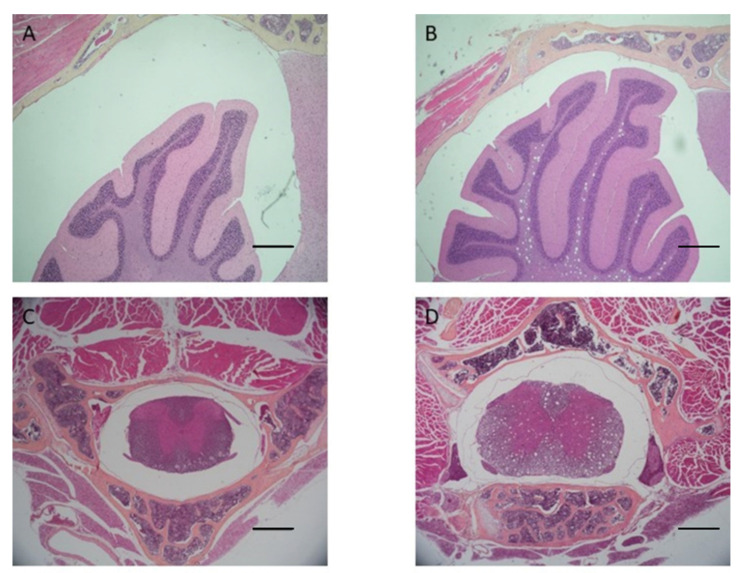
(**A**,**B**) Mouse head decalcified with EDTA-MW and Anna Morse’s solution, respectively, 40× scale bar 400 µm H&E staining. The formic acid solution causes some damage in the inner part of the cerebellum, identifiable as holes in the nervous tissue. (**C**,**D**) Mouse vertebra decalcified, respectively, with EDTA-MW and Anna Morse’s solution, 40× scale bar 400 µm H&E staining. A similar type of damage sustained by the cerebellum could be found in the spinal cord of the sample decalcified with Anna Morse’s solution. (**E**,**F**) Details taken at 100× of mouse vertebra decalcified with EDTA-MW and Anna Morse’solution, 400× scale bar 100 µm H&E staining. (**G**,**H**) Detail of bone marrow, hematopoietic tissue is better preserved in the sample decalcified with EDTA-MA (**G**) than in the one decalcified with Anna Morse’s solution (**H**) 400× scale bar 40 µm H&E staining.

## Data Availability

Not applicable.

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
