# Peer review of "Arduino Automated Microwave Oven for Tissue Decalcification"

_bioengineering, 2023, doi:10.3390/bioengineering10010079_

Round 1
Reviewer 1 Report
The initial premise of this work which states: "Hard tissues, like bone and teeth, must undergo a decalcification process to obtain a histological section" is not true. Hard tissues can and should be studied in non-decalcified sections. Specific methods have been developed for this purpose. Full details can be found in the following monographic book:
Dickson, G. R. (1984). Methods of calcified tissue preparation.
And in particular chapter 1 is highly recommended.
Schenk, R. K., Olah, A. J., & Herrmann, W. (1984). Methods of calcified tissue preparation. Preparation of calcified tissues for light microscopy, 1-56.
For hard tissue researchers the mineral is a very revealing source of information and the mineral itself is an element for imaging. It is a mistake to decalcify and waste more than half of the tissue.
If you want to decalcify hard tissue it is better to do it in tissue sections that have been previously embedded in plastic resins, post-embedding decalcification method. In this way the greatest amount of organic matrix is retained (Bonucci, E., & Reurink, J. (1978). The fine structure of decalcified cartilage and bone: a comparison between decalcification procedures performed before and after embedding. Calcified tissue research, 25(1), 179-190).
However, decalcification prior to paraffin embedding is also used as a routine method in hospital pathology laboratories for diagnostic purposes. For this purpose the specimens are decalcified at room temperature in acid solutions (e.g., with formic acid,) or in EDTA. In the last twenty years the decalcification process has been shortened by the use of commercially available laboratory microwaves. Their cost is around 40,000 €, but this is not prohibitive given the many applications that a laboratory microwave has for fixation, embedding, staining and decalcification.
The authors conclude that "Finally, with this setup is possible to spare a considerable amount of resources still having an instrument with similar capability as an expensive lab counterpart". However, with the suggested modification, the current microwave works only for 240 seconds per day and the decalcification times obtained are still very long. Nothing to do with the results obtained with a laboratory microwave with which the process is reduced from days to hours.
In conclusion, the work presented is a failed test.
Specific comments.
Line 92. Power must be set at maximum value.
Power in MW must be specified.
Lines 124-125. and verify the hypothesis that microwave irradiation can accelerate tooth decalcification
That microwave irradiation accelerates decalcification is a fact that has been proven for many years. Moreover, laboratory microwaves and specific protocols for their use in histology laboratories are commercially available.
Line 128. 14% EDTA disodium salt at pH=10 was chosen
Phosphate or Tris buffered EDTA at neutral pH is recommended.
Lines 132-133. The control group included samples decalcified in AM solution, and the test group, 132 included samples decalcified with EDTA in the microwave oven (EDTA-MW).
The control group should have been with EDTA solution if the efficacy of the proposed method in reducing the time required for complete decalcification is to be tested.
Lines 134. 3 sections of cow molar teeth of the same weight of 2 grams
In the processing of samples in a histology laboratory it is unusual to weigh the samples unless they are to be used for analysis of major mineral elements (Ca, P, Mg) and trace elements because to calculate their concentration we need to know the apparent mineral density.
It is better to refer to the dimensions of the sample (length and width) and especially its thickness. The experiment should have used tooth slices with different thicknesses, from 2 to 10 mm, to test the effectiveness of the proposed method.
Lines 139-140. The sample-liquid ratio has to be 1:30 in volume
A ratio of 1:100 in volume is recommended.
Line 160-161. The staining procedure is described in Table 2.
Detailing the staining procedure in Table 2 is completely unnecessary, it is routine. Delete the Table.
Lines 170-174. Cow's teeth in AM solution took 15 days to complete decalcification, whereas mouse head and vertebrae took 5 days. Cow's teeth in EDTA-MW took 25 days, whereas mouse head and vertebrae took 14 days. AM solution is faster than EDTA, but the microwave can speed up the decalcification process with the chelating agent from months to weeks.
The decalcification time in EDTA-MW is very long, days, which is similar to the time spent in conventional decalcification methods. Using a laboratory microwave the times are shortened from days to hours.
Reviewer 2 Report
The purpose of this study was to evaluate if a modified commercial microwave with an Arduino automation device can be used for hard tissue decalcification. Biological samples of cow molar teeth and mouse vertebrae and mouse heads were used. The samples were decalcified, sectioned, stained with H&E and were evaluated under the microscope. The samples decalcified using the Arduino automated microwave method were compared to samples decalcified with Morse's solution (formic acid).
The authors concluded that the suggested microwave method can be helpful.
At the introduction and discussion, the problems with using acids for decalcification of hard tissues were metnioned and discussed. It would have been useful and relevant for this study to include controls that were decalcified in chelating agent (EDTA) without the use of microwave, especially for the tissues that showed changes/poorer quality. In such a way, one could say something about the time it takes. I assume that decalcification in EDTA alone would preserve teh architecture of mouse vertebrae and heads. If the time needed is not reduced with the microwave technique, then one may question the use of a microwave...
Comments/suggestions: add bars in all microscopy photographs.
